# General medical publications during COVID-19 show increased dissemination despite lower validation

Nan Gai[1], Kazuyoshi Aoyama[1,2], David Faraoni[1], Neil M. Goldenberg[1,3], David N. Levin[1], Jason T. Maynes[1,4], Mark J. McVey[1,5], Farrukh Munshey[1], Asad Siddiqui[1], Timothy Switzer[1], Benjamin E. Steinberg[1,6]*

1 Department of Anesthesia and Pain Medicine, The Hospital for Sick Children, Toronto, Ontario, Canada,
2 Program in Child Health Evaluative Sciences, SickKids Research Institute, Toronto, Ontario, Canada,
3 Program in Cell Biology, SickKids Research Institute, Toronto, Ontario, Canada, 4 Program in Molecular Medicine, SickKids Research Institute, Toronto, Ontario, Canada, 5 Program in Translational Medicine, SickKids Research Institute, Toronto, Ontario, Canada, 6 Program in Neuroscience and Mental Health, SickKids Research Institute, Toronto, Ontario, Canada

* benjamin.steinberg@sickkids.ca

**Data Availability Statement:** All relevant data are within the paper and its Supporting Information files.

## Abstract

### Background

The COVID-19 pandemic has yielded an unprecedented quantity of new publications, contributing to an overwhelming quantity of information and leading to the rapid dissemination of less stringently validated information. Yet, a formal analysis of how the medical literature has changed during the pandemic is lacking. In this analysis, we aimed to quantify how scientific publications changed at the outset of the COVID-19 pandemic.

### Methods

We performed a cross-sectional bibliometric study of published studies in four high-impact medical journals to identify differences in the characteristics of COVID-19 related publications compared to non-pandemic studies. Original investigations related to SARS-CoV-2 and COVID-19 published in March and April 2020 were identified and compared to non-COVID-19 research publications over the same two-month period in 2019 and 2020. Extracted data included publication characteristics, study characteristics, author characteristics, and impact metrics. Our primary measure was principal component analysis (PCA) of publication characteristics and impact metrics across groups.

### Results

We identified 402 publications that met inclusion criteria: 76 were related to COVID-19; 154 and 172 were non-COVID publications over the same period in 2020 and 2019, respectively. PCA utilizing the collected bibliometric data revealed segregation of the COVID-19 literature subset from both groups of non-COVID literature (2019 and 2020). COVID-19 publications were more likely to describe prospective observational (31.6%) or case series (41.8%) studies without industry funding as compared with non-COVID articles, which were represented

**Funding:** Funding was provided through departmental funds from the Department of Anesthesia and Pain Medicine at the Hospital for Sick Children.

**Competing interests:** The authors have declared that no competing interests exist.

primarily by randomized controlled trials (32.5% and 36.6% in the non-COVID literature from 2020 and 2019, respectively).

## Conclusions

In this cross-sectional study of publications in four general medical journals, COVID-related articles were significantly different from non-COVID articles based on article characteristics and impact metrics. COVID-related studies were generally shorter articles reporting observational studies with less literature cited and fewer study sites, suggestive of more limited scientific support. They nevertheless had much higher dissemination.

## Introduction

The coronavirus disease 2019 (COVID-19) pandemic has given rise to an unprecedented quantity of publications in a short period of time as researchers worldwide attempt to report their experiences to better understand this new disease and identify promising treatments [1]. This has contributed to a COVID-19 "infodemic"–an overwhelming quantity of information, leading to the rapid dissemination of less stringently validated information [2].

Given the devastating severity of COVID-19, there is an understandable urgency to disseminate new findings. However, the rush to publish has potentially led to the compromise of scientific integrity [3]. This has led to advocacy for quality over quantity, cautioning that a crisis is no excuse for lowering scientific standards [3–5]. Yet, the COVID-19 pandemic has magnified traditional problems of "uninformative" clinical trials–those whose results are not useful to patients, clinicians, researchers, or policy makers [6, 7].

While specific concerns about COVID-19-related publications have been expressed [8], a formal analysis of the extent to which the medical literature has shifted during the pandemic is lacking. In this analysis, we aimed to quantify how scientific publications changed at the outset of the COVID-19 pandemic by performing a cross-sectional bibliometric study of published studies in four high-impact medical journals to identify differences in the characteristics of COVID-19 related publications compared to non-pandemic related studies.

## Methods

This is a cross-sectional bibliometric study of original COVID-19 related research publications in the four general medical journals with the highest impact factors [9]–*The Journal of the American Medical Association* (*JAMA*), *New England Journal of Medicine* (*NEJM*), *The Lancet*, and *Nature Medicine*. This study followed the Strengthening the Reporting of Observational Studies in Epidemiology (STROBE) reporting guidelines [10].

We searched for original investigations related to SARS-CoV-2 and COVID-19 published in March and April 2020 through MEDLINE. MEDLINE alone was used because it contained entries for all publications within our four journals of interest. Accordingly, other databases were not consulted. As comparison groups, we retrieved all non-COVID-19 research publications over the same two-month period in 2019 and 2020. We included original scientific research, and excluded opinion, news, and educational pieces. Two reviewers verified studies for inclusion and two reviewers audited extracted data. Any discrepancies in eligibility assessment and data collection were resolved by consensus. Extracted data included publication characteristics, study characteristics, author characteristics, and impact metrics. Impact

metrics (numbers of reads, citations, and tweets) were not normalized to the time since publication.

Categorical data are presented as counts and percentages and continuous data as medians and interquartile ranges (IQRs). Our primary measure was principal component analysis (PCA) of publication characteristics and impact metrics across groups. In our study, we sought to discover any differences in multiple article metrics between the 2020 COVID period and historical controls. Principal component analysis allows for the determination of the largest contributors to the variance in the data across all article metrics, in an unsupervised fashion without biasing data segregation [11]. Using PCA allows us to identify the most important features that capture the maximum information about the dataset, reducing dimensionality without any significant loss of information. Comparisons between groups were conducted using Chi-square or Fisher's exact tests for proportions and non-parametric Kruskal-Wallis tests with Dunn's multiple comparison for continuous data. Data for each journal were aggregated for analysis. *P* values less than 0.05 were considered statistically significant. Analyses were performed using GraphPad PRISM software version 7.0 and RStudio version 1.3.1056.

## Results

The initial MEDLINE literature search identified 1,119 total articles for consideration (262 COVID-related). We identified 402 publications that met inclusion criteria: 76 were related to COVID-19; 154 and 172 were non-COVID publications over the same period in 2020 and 2019, respectively (data available in S1 Dataset). Principal component analysis utilizing the collected bibliometric data revealed segregation of the COVID-19 literature subset from both groups of non-COVID literature (2019 and 2020), verifying that the bibliometric characteristics capture a change in publication metrics (Fig 1). The most significant contributions to the PCA came from metrics representing article dissemination (reads, tweets, and citations with 57%, 54%, and 43% each towards the first principal component, PC1). The two non-COVID subsets of data possess a near overlap in the PCA, indicating a strong consistency between the two years analyzed and emphasizing the uniqueness of the COVID-related literature.

To further evaluate how the published COVID-19 research literature differed from non-COVID-19 investigations, we first compared their publication characteristics (Table 1). Publication characteristics segregated by individual journal are provided in the Table in S1 Table. COVID-19 publications were more likely to describe prospective observational (31.6%) or case series (41.8%) studies without industry funding as compared with non-COVID articles, which were represented primarily by randomized controlled trials (32.5% and 36.6% in the non-COVID literature from 2020 and 2019, respectively). Moreover, COVID-related publications had lower word counts with fewer citations of other medical literature. While the number of authors was unchanged, the number of author affiliations was decreased, suggesting a lower level of collaborative or multi-institutional studies. There was no observed difference in the proportion of female first or corresponding authors. For *Nature Medicine*, the only evaluated journal to report submission dates, COVID-related submissions were published in a much shorter amount of time (35.1 days versus 288.3 and 305.3 days for 2020 and 2019 non-COVID publications, respectively).

The observed differences in publication characteristics presumably represents the initial effort to quickly provide clinicians and policymakers with information in the early phase of the pandemic, regardless of quality. To objectively evaluate the extent to which the COVID-19 literature was disseminated, we analyzed the number of accesses, tweets, and citations within our bibliometric dataset. Publications related to COVID had an order of magnitude greater accesses, tweets, and citations compared with non-COVID publications from the same period

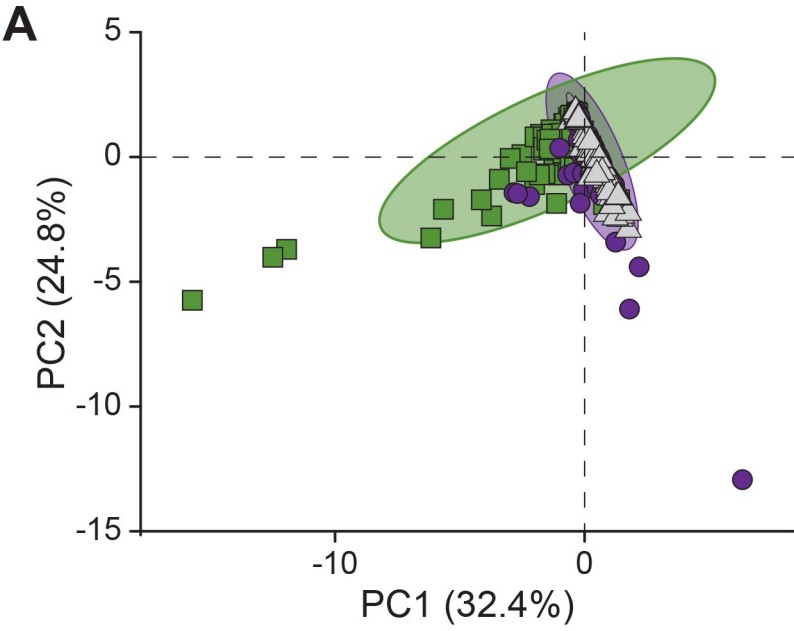

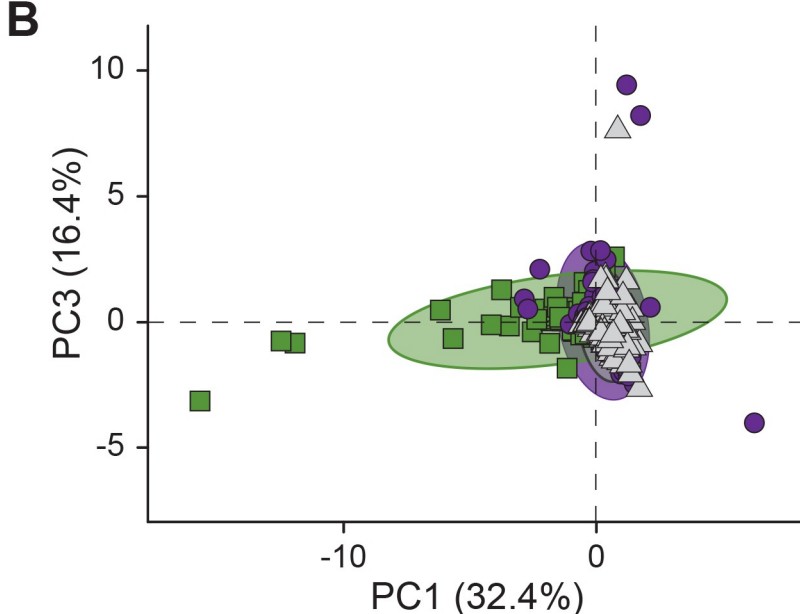

**Fig 1. Principal component analysis of COVID and non-COVID publication characteristics and impact metrics.**
Each point in the plot corresponds to a single characteristic provided in Table 1 for COVID (green square) and non-COVID publications from 2019 (purple circle) and 2020 (gray triangle). Principal component 1 (PC1) is shown plotted against (A) PC2 and (B) PC3. PC1, PC2, and PC3 respectively account for 32.4%, 24.8% and 16.4% of the variability. Non-COVID publications from 2019 and 2020 clusters overlap, whereas COVID publications cluster separately. This

unbiased analysis suggests COVID-related publications differ from both concurrent and historic non-COVID publications.

in both 2019 and 2020 (Table 1). This absolute difference does not consider the greater time since publication of articles from 2019 and therefore may conservatively underestimate the unparalleled rate at which observational data spread across the international medical community.

## Discussion

Using an unbiased approach, our PCA suggests that published pandemic-related studies have different article characteristics and impact metrics compared with non-COVID studies. They

**Table 1. Publication characteristics and impact.**

| | Non-COVID publications | | COVID publications | P value [a] | |
| --- | --- | --- | --- | --- | --- |
| | **2019** | **2020** | | *COVID vs non-COVID 2019* | *COVID vs non-COVID 2020* |
| **Articles (n)** | 172 | 154 | 76 | | |
| **Article type, No. (%)** | | | | | |
| Meta-analysis | 6 (3.5) | 4 (2.6) | 0 (0) | <0.0001 | <0.0001 |
| Systematic review | 4 (2.3) | 6 (3.9) | 2 (2.6) | | |
| Narrative review | 17 (9.9) | 16 (10.4) | 4 (5.3) | | |
| RCT | 63 (36.6) | 50 (32.5) | 1 (1.3) | | |
| Cohort / prospective | 30 (17.4) | 29 (18.8) | 24 (31.6) | | |
| Case-control | 3 (1.7) | 4 (2.6) | 2 (2.6) | | |
| Case report or series | 14 (8.1) | 17 (11.0) | 31 (40.8) | | |
| Basic biomedical research / preclinical | 18 (10.5) | 18 (11.7) | 5 (6.6) | | |
| Other | 17 (9.9) | 10 (6.5) | 7 (9.2) | | |
| **Study characteristics [b], No. (%)** | | | | | |
| Registered trial | 75 (47.5) | 54 (35.5) | 0 (0) | <0.0001 | <0.0001 |
| Industry funding | 37 (22.2) | 48 (31.2) | 2 (2.7) | <0.0001 | <0.0001 |
| **Publication characteristics** | | | | | |
| Author number, median (IQR) | 15 (17) | 12 (16) | 10.5 (12.75) | 0.4499 | 1 |
| Author affiliations, median (IQR) | 8 (7) | 7 (13) | 4 (4) | <0.0001 | <0.0001 |
| Female corresponding or first author [b], No. (%) | 59 (36.4) | 56 (36.8) | 24 (33.8) | 0.7671 | 0.7646 |
| Time to publication, days, mean (SD) [c] | 305.3 (124.2) | 288.3 (99.7) | 35.1 (4.6) | <0.0001 | 0.0001 |
| Word count, median (IQR) | 3816 (2063) | 3746 (2061) | 914 (2139) | <0.0001 | <0.0001 |
| References, median (IQR) | 33 (19.75) | 33 (24.5) | 6 (22) | <0.0001 | <0.0001 |
| **Publication impact [d], median (IQR)** | | | | | |
| Reads [e] | 17 648 (21 959) | 9 652 (15 110) | 224 714 (389 243) | <0.0001 | <0.0001 |
| Tweets | 168.5 (250.5) | 81.5 (149.5) | 1202 (4014 | <0.0001 | <0.0001 |
| Times cited | 25 (33.75) | 2 (4) | 50.5 (125.3) | 0.1414 | <0.0001 |

Abbreviations: COVID, Coronavirus Disease; *JAMA*, Journal of the American Medical Association; *NEJM*, New England Journal of Medicine; RCT, Randomized Controlled Trial.

[a] *P* values, adjusted for multiple comparison, shown for comparison between COVID and non-COVID publications from the indicated year.

[b] Articles in which study characteristic was not reported or in which gender of author was unknown were excluded from calculation of the proportion.

[c] Includes only *Nature Medicine* publications as submission dates not reported for *JAMA*, *The Lancet*, or *NEJM*.

[d] Reads, tweets and times cited are reported as absolute numbers and are not normalized to their time since publication.

[e] Excludes articles published in *The Lancet*, which does not list article reads as part of their Altmetrics.

generally consist of shorter articles reporting observational studies with less literature cited and fewer study sites, suggestive of more limited scientific support. Yet, pandemic-related research is associated with greater reach in terms of readership, citations, and tweets, which speaks to the strong appetite for pandemic-related findings.

The publication characteristics described in our analysis reflect the urgency with which the medical, scientific, and lay communities sought information as the pandemic evolved. This on-going need, however, should be tempered with scientific and ethical oversight that is at least as rigorous as normal times with a focus on well-designed trials and not rapid dissemination of low-quality data. The potential harms of producing multiple iterations of lower-quality studies have been identified, including wasting of resources, lapses in the ethical standard of scientific reporting, delaying the conduct of higher-level evidence trials, diluting the quality of available evidence, and endangering the ethical responsibility to patients who enroll in trials with the expectation of assisting in medical and scientific advancement [6, 12, 13]. Researchers should endeavour to maintain high-quality research methods by increasing collaboration across multiple centres, helping to overcome limitations that may exist from single-centre efforts [3, 14]. International teams working in concert and not in competition on well-designed studies would greatly improve the capacity to detect clinically meaningful effects to inform the international health system's efforts against COVID-19. For example, research consortia could establish research priorities and promote the implementation of master protocols with adaptive platforms [15–17]. This type of approach is designed for the perpetual investigation of multiple interventions with timely adaptation, an ideal framework for our evolving COVID-19 health crisis that would facilitate wider collaboration and mitigate against the production of low-quality evidence and poor scientific reporting.

Efforts have also focused on the expanding COVID-19 literature itself using both manual and automated methods. Content experts have been vetting the published literature to provide health care workers and policymakers with curated digital compendiums of high-quality research papers, such as the 2019 Novel Coronavirus Research Compendium [18]. Computational approaches are being used to mine the published COVID-19 literature to answer key questions related to the pandemic [19]. As these resources continue to grow, increasing effort will be required to ensure that the medical, scientific, and lay communities can engage with the resulting data and analyses in a meaningful way.

Our analysis, however, has limitations. We focus on the earliest phase of pandemic in order to capture how the medical community first pivoted to acquire and disseminate COVID-19-related knowledge. This potentially biases our results towards observational studies as there would be limited time to advance and report more rigorous study designs, such as randomized controlled trials. Moreover, to efficiently disseminate medical knowledge, the included journals made pandemic-related content freely available, which may have contributed to the observed increase in impact metrics. Lastly, our bibliometric analysis does not consider the root cause of the disparity between COVID and non-COVID publications. This is likely multifactorial but could, in part, reflect the feasibility of a timely study completion, variable adherence to reporting standards, and a strained peer review system. Ongoing evaluations of the publication process over the entirety of the pandemic will inform how the scientific community can most effectively, safely, and ethically disseminate valuable medical knowledge in a time of acute crisis.

## Conclusion

COVID-19 led to a significant change in the characteristics of research studies across high-impact general medical journals. During this pandemic, the rapid and broad dissemination of

research findings, regardless of underlying quality, were amplified and potentially contributed to the infodemic of misinformation at a time when best evidence needs to be emphasized. Ultimately, relaxing the rigorous standards for scientific research, although tempting for many altruistic reasons during a pandemic, may not actually achieve the objective of producing a solid evidence-based foundation upon which patients, clinicians, and policymakers can make meaningful decisions. The scientific and medical communities must strongly advocate for the thoughtful selection of high-quality research that will ensure the generation of meaningful knowledge and that participants of scientific trials who volunteer their health experience do not do so in vain.

## Supporting information

**S1 Dataset. The dataset used for the analyses in this study.**
(XLSX)

**S1 Table. Publication characteristics and impact by journal.**
(DOCX)

## Author Contributions

**Conceptualization:** Nan Gai, David Faraoni, Jason T. Maynes, Benjamin E. Steinberg.

**Data curation:** Nan Gai, Kazuyoshi Aoyama, David Faraoni, Neil M. Goldenberg, David N. Levin, Jason T. Maynes, Mark J. McVey, Farrukh Munshey, Asad Siddiqui, Timothy Switzer, Benjamin E. Steinberg.

**Formal analysis:** Nan Gai, Kazuyoshi Aoyama, David Faraoni, Neil M. Goldenberg, David N. Levin, Jason T. Maynes, Mark J. McVey, Farrukh Munshey, Asad Siddiqui, Timothy Switzer, Benjamin E. Steinberg.

**Methodology:** Nan Gai, Kazuyoshi Aoyama, David Faraoni, Neil M. Goldenberg, David N. Levin, Jason T. Maynes, Mark J. McVey, Farrukh Munshey, Asad Siddiqui, Timothy Switzer, Benjamin E. Steinberg.

**Project administration:** Benjamin E. Steinberg.

**Resources:** Benjamin E. Steinberg.

**Supervision:** Benjamin E. Steinberg.

**Validation:** Benjamin E. Steinberg.

**Visualization:** Jason T. Maynes.

**Writing – original draft:** Nan Gai, Benjamin E. Steinberg.

**Writing – review & editing:** Nan Gai, Kazuyoshi Aoyama, David Faraoni, Neil M. Goldenberg, David N. Levin, Jason T. Maynes, Mark J. McVey, Farrukh Munshey, Asad Siddiqui, Timothy Switzer, Benjamin E. Steinberg.

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
