## [Decision Letter · Decision Letter 0]

4 Dec 2020

PONE-D-20-29109

General Medical Publications During COVID-19 Show Increased Dissemination Despite Lower Validation.

PLOS ONE

Dear Dr. Steinberg,

Two reviewers submitted their reviews and their recommendations do not exactly match.  I wish to say that the topic is important.  Indeed, the epidemic has opened doors for anyone wishing to publish anything that has to do with COVID-19.  However, the comments by one of the reviewers is critical.  The authors need to justify the methodology used.  I am not a statistician.  I know PCA can be used to identify subjects with similar attributes.  Whether dispersion in PCA proves that COVID-19 publications are of inferior quality is a question readers will ask. Reviewer no. 2 projected this problem - is PCA the right tool to evaluate the study question?

Due to the importance of this subject, after careful consideration, I feel that it has merit but does not fully meet PLOS ONE’s publication criteria as it currently stands. Therefore, I invite you to submit a revised version of the manuscript that addresses the points raised during the review process.

We look forward to receiving your revised manuscript.

Kind regards,

Itamar Ashkenazi

Academic Editor

PLOS ONE

Journal Requirements:

2. Thank you for stating the following in the Funding Statement Section of your manuscript:

"Funding was provided through departmental funds from the Department of Anesthesia and Pain Medicine at the Hospital for Sick Children."

"The authors received no specific funding for this work."

Reviewers' comments:

Reviewer's Responses to Questions

**Comments to the Author**

1. Is the manuscript technically sound, and do the data support the conclusions?

Reviewer #1: Yes

Reviewer #2: Partly

2. Has the statistical analysis been performed appropriately and rigorously? 

Reviewer #1: Yes

Reviewer #2: No

3. Have the authors made all data underlying the findings in their manuscript fully available?

Reviewer #1: Yes

Reviewer #2: Yes

4. Is the manuscript presented in an intelligible fashion and written in standard English?

Reviewer #1: Yes

Reviewer #2: Yes

5. Review Comments to the Author

Reviewer #1: During COVID-19, many articles published in various journals. However, I noticed that many authors are not related to medical field!! Many authors want to gain more citations by choosing COVID-19 topic. Also, many journals are publishing COVID-19 related publications without much review (rapid review) so, sometimes quality of the manuscript is poor but dissemination (reach) is greater as non-medical field persons are searching for COVID-19 related articles.

This manuscript has novelty and timely to analyze the COVID-19 and non-COVID-19 related articles characteristics.

Manuscript is well written and it can be published with minor corrections.

Line-2: There should not be period (full stop) in title.

Line-60: MEDLINE is the pioneer database but author should have to describe for choosing this database and excluding other databases.

Line-62: Single reviewer

Reviewer #2: I will focus on methods and reporting

Major

1) Abstract is quite short and poor. no information on methods or results.

2) The methods are unclear, especially the role of PCA and what the outcomes are e.g. impact metrics.

3) Why don't the authors use established quality assessment tools for observational studies and report that instead?

4) I don't follow why the data were aggregated within each journal and each study was not included as a different case.

Minor

1) Rstudio is the shell, reference what's under the hood i.e. R version.

6. PLOS authors have the option to publish the peer review history of their article (what does this mean?). If published, this will include your full peer review and any attached files.

Reviewer #1: **Yes: **Anjum Sherasiya

Reviewer #2: No

---

## [Author Response · Author response to Decision Letter 0]

28 Dec 2020

RE: PONE-D-20-29109

General Medical Publications During COVID-19 Show Increased Dissemination Despite Lower Validation.

PLOS ONE

Dear Dr. Ashkenazi:

We would like to thank you and the expert reviewers for the thoughtful evaluation of our paper and helpful comments. Having addressed and incorporated the reviewers’ suggestions, we now submit a much improved manuscript for your consideration. In addition, our revised manuscript incorporates the requested changes to conform to your journal’s requirements. Please find attached below detailed point-by-point responses to the reviewers’ concerns. 

Thank you again to you and the reviewers.

Sincerely,

Benjamin Steinberg, on behalf of all co-authors

Response to Dr. Ashkenazi, Academic Editor

(1) The authors need to justify the methodology used. I am not a statistician. I know PCA can be used to identify subjects with similar attributes. Whether dispersion in PCA proves that COVID-19 publications are of inferior quality is a question readers will ask. Reviewer no. 2 projected this problem - is PCA the right tool to evaluate the study question?

Thank you for this insightful and generally positive review. We agree that a greater justification of our use of PCA is needed, which we provide in our revised manuscript. This is described in greater detail below in response to Reviewer #2’s specific query.

In brief, we sought to discover any differences in multiple article metrics between the 2020 COVID period and historical controls. Principal component analysis allows for the determination of the largest contributors to the variance in the data across all article metrics, in an unsupervised fashion without biasing data segregation [Jolliffe IT and Cadima J. (2016) Principal component analysis: a review and recent developments. Phil. Trans. R. Soc. A. 374, 20150202]. Importantly, in our cross-sectional study of publications in four general medical journals, COVID-related articles were significantly different from non-COVID articles based on article characteristics and impact metrics. While it is difficult to ascertain quality per se, we describe how COVID-related studies were generally shorter articles reporting observational studies with less literature cited and fewer study sites, suggestive of more limited scientific support. We discuss these findings in our Discussion section.

Response to Dr. Sherasiya, Reviewer #1

(1) This manuscript has novelty and timely to analyze the COVID-19 and non-COVID-19 related articles characteristics… Manuscript is well written and it can be published with minor corrections.

Thank you for your kind and thoughtful review. We have updated our manuscript with your suggested corrections as listed below.

(2) Line-2: There should not be period (full stop) in title.

The period in the title has been removed.

(3) Line-60: MEDLINE is the pioneer database but author should have to describe for choosing this database and excluding other databases.

Thank you for bringing up this important point of clarification. As you note, MEDLINE is a pioneering database, commonly used for bibliometric studies. We chose MEDLINE specifically to facilitate our study identification and data extraction because it is easy to use and contained entries for all publications within our four journals of interest. It was not necessary to consult other databases because our MEDLINE search captured all potential studies for inclusion. 

A broader search, not limited to specific journals would have necessitated inclusion of other repositories such as Embase, Cochrane Controlled Clinical Trial Register, PubMed, and CINAHL. That, however, was not the case with our investigation.

Our revised manuscript has been updated to include a point of clarification and justification of our choice of MEDLINE over other databases as follows:

“We searched for original investigations related to SARS-CoV-2 and COVID-19 published in March and April 2020 through MEDLINE. MEDLINE alone was used because it contained entries for all publications within our four journals of interest. Accordingly, other databases were not consulted.”

(4) Line-62: Single reviewer

Thank you for alerting us to this error. In fact, two reviewers verified each study for inclusion. We have corrected this error in our revised manuscript:

“Two reviewers verified studies for inclusion and two reviewers audited extracted data.”

Response to Reviewer #2

1) Abstract is quite short and poor. no information on methods or results.

We acknowledge that the Abstract in our initial submission did not effectively convey detail on our methods and results. We have since expanded our Abstract to include a more detailed description of our methods and results in our revised submission. The revisions are as follows:

Background: The COVID-19 pandemic has yielded an unprecedented quantity of new publications, contributing to an overwhelming quantity of information and leading to the rapid dissemination of less stringently validated information. Yet, a formal analysis of how the medical literature has changed during the pandemic is lacking. In this analysis, we aimed to quantify how scientific publications changed at the outset of the COVID-19 pandemic. 

Methods: We performed a cross-sectional bibliometric study of published studies in four high-impact medical journals to identify differences in the characteristics of COVID-19 related publications compared to non-pandemic related studies. Original investigations related to SARS-CoV-2 and COVID-19 published in March and April 2020 were identified and compared to non-COVID-19 research publications over the same two-month period in 2019 and 2020. Extracted data included publication characteristics, study characteristics, author characteristics, and impact metrics. Our primary measure was principal component analysis (PCA) of publication characteristics and impact metrics across groups.

Results: We identified 402 publications that met inclusion criteria: 76 were related to COVID-19; 154 and 172 were non-COVID publications over the same period in 2020 and 2019, respectively. PCA utilizing the collected bibliometric data revealed segregation of the COVID-19 literature subset from both groups of non-COVID literature (2019 and 2020). COVID-19 publications were more likely to describe prospective observational (31.6%) or case series (41.8%) studies without industry funding as compared with non-COVID articles, which were represented primarily by randomized controlled trials (32.5% and 36.6% in the non-COVID literature from 2020 and 2019, respectively).

Conclusion: In this cross-sectional study of publications in four general medical journals, COVID-related articles were significantly different from non-COVID articles based on article characteristics and impact metrics. COVID-related studies were generally shorter articles reporting observational studies with less literature cited and fewer study sites, suggestive of more limited scientific support. They nevertheless had much higher dissemination.

2) The methods are unclear, especially the role of PCA and what the outcomes are e.g. impact metrics.

We agree that our initial submission did not adequately describe our study’s full methodology. As described below, our revised manuscript now includes an expanded Methods section that specifically describes the role and rationale of PCA, as well as delineates each of our impact metrics. 

In our study, we sought to discover any differences in multiple article metrics between the 2020 COVID period and historical controls. Principal component analysis allowed us to determine the largest contributors to the variance in the data across all article metrics, in an unsupervised fashion without biasing data segregation (Jolliffe, I. T., and Cadima, J. (2016) Principal component analysis: a review and recent developments. Phil. Trans. R. Soc. A. 374, 20150202). We found the most important features that capture the maximum information about the dataset, reducing dimensionality without any significant loss of information. Since the principal components are independent, we eliminate correlated features, important in our data set as there was a chance that quality metrics could be tied. Were this the case, validating a lack of correlation would have been challenging and time consuming for this dataset using traditional statistical methods. Additional advantages to PCA are that it reduces our chance of overfitting the data and overemphasizing the change in certain article metrics. In these ways, PCA was well suited for the type of data and analysis we were performing, revealing distinct differences between the article categories without prior assumptions or bias while still considering all collected metrics. We have tried other unsupervised methods (i.e. partial least square discriminant analysis and sparse PLA-DA) and found similar results. Lastly, PCA has been used in other similar studies (e.g. Bollen, J., Van de Sompel, H., Hagberg, A., and Chute, R. (2009) A Principal Component Analysis of 39 Scientific Impact Measures. PLoS ONE. 4, e6022).

We have also clarified the impact metrics extracted for analysis: number of reads, number of citations, and number of tweets. 

The relevant additions to our revised manuscript are as follows:

“Our primary measure was principal component analysis (PCA) of publication characteristics and impact metrics across groups. In our study, we sought to discover any differences in multiple article metrics between the 2020 COVID period and historical controls. Principal component analysis allows for the determination of the largest contributors to the variance in the data across all article metrics, in an unsupervised fashion without biasing data segregation [Jolliffe IT and Cadima J. (2016) Principal component analysis: a review and recent developments. Phil. Trans. R. Soc. A. 374, 20150202]. Comparisons between groups were conducted using Chi-square or Fisher’s exact tests for proportions and non-parametric Kruskal-Wallis tests with Dunn’s multiple comparison for continuous data. Data for each journal were aggregated for analysis.”

“Extracted data included publication characteristics, study characteristics, author characteristics, and impact metrics. Impact metrics (numbers of reads, citations, and tweets) were not normalized to the time since publication.”

3) Why don't the authors use established quality assessment tools for observational studies and report that instead?

This is an excellent suggestion that we had discussed when first designing our study. Specifically, we had considered using established checklists to assess for quality of publication. However, we chose not to include these tools as we were not assessing only observational studies. The resulting collection of studies that met our inclusion criteria were heterogeneous in nature (i.e. by study design) without the necessary sample sizes per type of study to perform a comparison within specific study design groups. Moreover, there is no validated method for comparing between different types of quality assessment tools, which would have been required given the different types of study designs within our inclusion criteria.

4) I don't follow why the data were aggregated within each journal and each study was not included as a different case.

Thank you for bringing attention to this important point of clarification. We did consider separating analyses by journal as well. Given that our included journals are all high-impact, general medical journals, we felt that further individual comparisons would not provide meaningful information when our primary objective was to look at overall changes in publication approach among all high-impact, general medical journals between the COVID and non-COVID literature. The main goal was to look at scientific publications in general, and not focus on any specific journal. Additionally, the small numbers of publications per journal adds a further layer of multiple comparisons that our study is not appropriately powered for. 

Nevertheless, we agree that our readership may be interested in the data at the individual journal level. As a result, we have included a supplemental table with the data segregated by each journal (see S1 Table). Moreover, our submission provides our entire dataset (available in S1 Dataset). The interested reader is therefore able to undertake a specific analysis of the data segregated by journal at their discretion. 

We have clarified this in the Methods and Results sections: 

“We identified 402 publications that met inclusion criteria: 76 were related to COVID-19; 154 and 172 were non-COVID publications over the same period in 2020 and 2019, respectively (data available in S1 Dataset).”

“To further evaluate how the published COVID-19 research literature differed from non-COVID-19 investigations, we first compared their publication characteristics (Table 1). Publication characteristics segregated by individual journal are provided in the Table in S1 Table.”

5) Rstudio is the shell, reference what's under the hood i.e. R version.

Thank you for identifying this oversight and potential source of confusion. The R version is included in our revised manuscript:

 “Analyses were performed using GraphPad PRISM software version 7.0 and RStudio version 1.3.1056.”

Journal Requirements: When submitting your revision, we need you to address these additional requirements.

Our revised manuscript now meets the PLOS ONE style requirements, including those for file naming.

2. Please include your amended [funding] statements within your cover letter; we will change the online submission form on your behalf.

Thank you for identifying this oversight. Funding-related text has been removed from the manuscript. 

3. We note that you have indicated that data from this study are available upon request. PLOS only allows data to be available upon request if there are legal or ethical restrictions on sharing data publicly… We will update your Data Availability statement on your behalf to reflect the information you provide.

Thank you for this reminder about data reporting. As outlined in our revised cover letter, there are no restriction on sharing our dataset. Accordingly, we have uploaded our dataset as a supplementary dataset (S1). We have added a reference to this supplementary information in the Results section:

“We identified 402 publications that met inclusion criteria: 76 were related to COVID-19; 154 and 172 were non-COVID publications over the same period in 2020 and 2019, respectively (data available in S1 Dataset).”

---

## [Decision Letter · Decision Letter 1]

20 Jan 2021

General Medical Publications During COVID-19 Show Increased Dissemination Despite Lower Validation.

PONE-D-20-29109R1

Dear Dr. Steinberg,

We’re pleased to inform you that your manuscript has been judged scientifically suitable for publication and will be formally accepted for publication once it meets all outstanding technical requirements.

Both one of the reviewers and I had some concerns with ACP being used as a major evaluation tool towards your endpoint.  However with the help of the two reviewers I have decided that it is up to the intelligent reader to decide whether ACP improves the analysis or not.  As you mention, the descriptive analysis provide sufficient information.

Kind regards,

Itamar Ashkenazi

Academic Editor

PLOS ONE

Additional Editor Comments (optional):

Reviewers' comments:

Reviewer's Responses to Questions

**Comments to the Author**

1. If the authors have adequately addressed your comments raised in a previous round of review and you feel that this manuscript is now acceptable for publication, you may indicate that here to bypass the “Comments to the Author” section, enter your conflict of interest statement in the “Confidential to Editor” section, and submit your "Accept" recommendation.

Reviewer #1: All comments have been addressed

Reviewer #2: All comments have been addressed

2. Is the manuscript technically sound, and do the data support the conclusions?

Reviewer #1: Yes

Reviewer #2: Partly

3. Has the statistical analysis been performed appropriately and rigorously? 

Reviewer #1: N/A

Reviewer #2: Yes

4. Have the authors made all data underlying the findings in their manuscript fully available?

Reviewer #1: Yes

Reviewer #2: Yes

5. Is the manuscript presented in an intelligible fashion and written in standard English?

Reviewer #1: Yes

Reviewer #2: Yes

6. Review Comments to the Author

Reviewer #1: You attended all the comments satisfactorily.

Reviewer #2: I am still not entirely convinced why the authors did not used established quality tools, and analyse by groups. That would make the analysis more valuable.

7. PLOS authors have the option to publish the peer review history of their article (what does this mean?). If published, this will include your full peer review and any attached files.

Reviewer #1: **Yes: **Anjum Sherasiya

Reviewer #2: No

---

## [Editor Report · Acceptance letter]

22 Jan 2021

PONE-D-20-29109R1 

General Medical Publications During COVID-19 Show Increased Dissemination Despite Lower Validation 

Dear Dr. Steinberg:

I'm pleased to inform you that your manuscript has been deemed suitable for publication in PLOS ONE. Congratulations! Your manuscript is now with our production department. 

Kind regards, 

on behalf of

Dr. Itamar Ashkenazi 

Academic Editor

PLOS ONE